# Anti-Dengue: A Machine Learning-Assisted Prediction of Small Molecule Antivirals against Dengue Virus and Implications in Drug Repurposing

**DOI:** 10.3390/v16010045

**Published:** 2023-12-27

**Authors:** Sakshi Gautam, Anamika Thakur, Akanksha Rajput, Manoj Kumar

**Affiliations:** 1Virology Unit, Institute of Microbial Technology, Council of Scientific and Industrial Research (CSIR), Sector 39A, Chandigarh 160036, India; gautamsakshi@imtech.res.in (S.G.); anamikathakur@imtech.res.in (A.T.); akanksharajput.bio@gmail.com (A.R.); 2Academy of Scientific and Innovative Research (AcSIR), Ghaziabad 201002, India

**Keywords:** dengue virus, machine learning, predictive models, QSAR, web server

## Abstract

Dengue outbreaks persist in global tropical regions, lacking approved antivirals, necessitating critical therapeutic development against the virus. In this context, we developed the “Anti-Dengue” algorithm that predicts dengue virus inhibitors using a quantitative structure–activity relationship (QSAR) and MLTs. Using the “DrugRepV” database, we extracted chemicals (small molecules) and repurposed drugs targeting the dengue virus with their corresponding IC_50_ values. Then, molecular descriptors and fingerprints were computed for these molecules using PaDEL software. Further, these molecules were split into training/testing and independent validation datasets. We developed regression-based predictive models employing 10-fold cross-validation using a variety of machine learning approaches, including SVM, ANN, kNN, and RF. The best predictive model yielded a *PCC* of 0.71 on the training/testing dataset and 0.81 on the independent validation dataset. The created model’s reliability and robustness were assessed using William’s plot, scatter plot, decoy set, and chemical clustering analyses. Predictive models were utilized to identify possible drug candidates that could be repurposed. We identified goserelin, gonadorelin, and nafarelin as potential repurposed drugs with high pIC50 values. “Anti-Dengue” may be beneficial in accelerating antiviral drug development against the dengue virus.

## 1. Introduction

Dengue, a viral disease transmitted by mosquitoes, exhibits a rapid transmission rate and is particularly common in tropical and subtropical areas. Consequently, it presents a substantial burden in terms of both mortality and morbidity [1]. Dengue was first registered in 1780 in Madras (now Chennai). The initial virology-confirmed outbreak occurred in Calcutta and along India’s eastern coast from 1963 to 1964. DHF was recorded in the Philippines in 1953–1954 [2]. Since 1950, frequent dengue outbreaks have occurred in Southeast Asian countries [3]. The World Health Organization (WHO) has reported a significant increase in the global burden of dengue over the past two decades. Roughly half of the world’s population is at risk of dengue infection, with an estimated 100 to 400 million infections yearly [4,5].

Dengue virus (DENV) is a single positive-stranded RNA virus belonging to the genus *Flavivirus* and *Flaviviridae* family. DENV has four serotypes, namely DENV-1, DENV-2, DENV-3, and DENV-4. DENV transmission occurs through the bite of DENV-carrying female *Aedes* mosquitoes, mainly by *Aedes aegypti* and, rarely, by *Aedes albopictus*, which leads to severe health issues known as dengue fever (DF), dengue hemorrhagic fever (DHF), and dengue shock syndrome (DSS) [5]. DENV comprises 10,723 nucleotides (approximately 11 kb), enciphering larger polyprotein precursors containing ~3391 amino acid residues. DENV polyproteins, after cleavage by host and virus proteases, constitute three structural proteins named C (capsid); prM (pre-membrane); E (envelope); and seven nonstructural proteins called NS1, NS2A, NS2B, NS3, NS4A, NS4B, and NS5 [6]. The clinical symptoms of dengue infection can range from asymptomatic to severe illnesses that may result in fatalities. Different categories exist in symptomatic cases, including mild acute undifferentiated febrile illness (UF), DF, DHF, DSS, and uncommon dengue (UD) or expanded dengue syndrome (EDS) [7].

Several research groups identified novel potent DENV inhibitors. Low, June Su Yin et al. identified Narasin as a novel antiviral agent with an IC_50_ of less than 1 μM against all DENV serotypes [8]. In another study, Raekiansyah, Muhareva et al. highlighted Brefeldin A as a promising and novel antiviral compound, displaying an IC_50_ range of 54.6 to 65.7 nM against all DENV serotypes [9]. Bardiot et al. studied the potential of KU Leuven’s compound library in inhibiting DENV-2 through a CPE reduction assay. They determined a promising inhibitor, 2-((3,4-dimethoxyphenyl) amino)-1-(1H-indol-3-yl)-2-phenylethan-1-one, for DENV [10].

Further, several experimental studies have been performed to determine the activity of repurposed drugs against the DENV. Drug repurposing could be a promising approach to looking for effective antivirals against the DENV. For example, quinine [11], N-Acetylcysteine [12], and Antiemetic Metoclopramide [13] have been used as repurposed drugs against DENV. Likewise, many more antivirals as potential repurposed drug candidates have been explored against the DENV [14]. Still, fewer antivirals are under clinical trial; therefore, we must explore more chemicals/inhibitors to get a highly effective and potent antiviral against DENV.

In this endeavor, computational approaches can be used to predict potent antivirals to reduce the time and cost. It could also be advantageous to accelerate the drug discovery process. In light of this, our group developed various machine learning-based antiviral predictors using the quantitative structure–activity relationship (QSAR) information of molecules/peptides such as AVCpred [15], AVPpred [16], AVP-IC50 Pred [17], HIVprotI [18], Anti-flavi [19], etc. Recently, we have developed a predictive algorithm for SARS-CoV-2, i.e., anti-corona [20], and for Ebola virus, i.e., anti-Ebola [21]. However, the platform is required to predict the repurposed drugs targeting the DENV utilizing machine learning techniques (MLTs).

In this study, we developed the “Anti-Dengue” predictive algorithm using various MLTs like support vector machine (SVM), artificial neural network (ANN), k-nearest neighbor (kNN), and random forest (RF). This algorithm predicts the efficacy of chemicals and drugs against DENV by assessing their inhibition efficiency, measured in terms of pIC50 and IC50 values (μM). Furthermore, we have also identified various effective repurposed drug candidates by scanning the “DrugBank” database through the best predictive model.

## 2. Materials and Methods

For developing the anti-dengue predictor, the workflow is given in Figure 1.

### 2.1. Data Collection

The antiviral entries were procured from the “DrugRepV” database to develop an “Anti-Dengue” predictor. The “DrugRepV” database encompasses chemicals (small molecules) and repurposed drugs designed to target epidemic and pandemic viruses, comprising a total of 8485 entries. This dataset provides comprehensive information, including antiviral names, drug types, primary and secondary indications, viral strains, pathways, assay details, clinical status, and more [22].

The steps for fetching out the antiviral entries are given below using the standard method [23]:We obtained 900 antiviral entries for the DENV in the “DrugRepV” database.The antiviral entries were filtered based on IC_50_/EC_50_ values, SMILES, molecular weight, etc. to acquire only relevant candidates, i.e., 238.Using the formula pIC_50_ = −log_10_(IC_50_(M)), the IC_50_ is converted into pIC_50_, where the IC_50_ is the dimensionless activity that can be expressed in molar concentrations. Higher values of pIC_50_ showed greater potency and vice versa.

The dataset containing drugs/inhibitors to create the model is given in Appendix A.

### 2.2. Descriptor Calculation

The chemical structures of antiviral candidates were used to procure the chemical information, such as the simplified molecular-input line-entry system (SMILES), then reformed into 3D-SDF format utilizing the open Babel v3.1.1 command line tool [24]. Further, these SDF files served as inputs for withdrawing chemical descriptors and fingerprints.

### 2.3. Compounds/Inhibitors Feature Extraction

The computation of 1D, 2D, and 3D molecular descriptors and fingerprints using 3D-SDF structures was performed using PaDEL software (version 2.21) to calculate 17,968 descriptors [25]. One-dimensional descriptors are substructural molecular fragment-based descriptors (H-Bond acceptor/donor, fingerprints, fragments count, etc.). Two-dimensional descriptors are structural and physicochemical properties-based descriptors (topological and electronic information, topological descriptors, connectivity indices, etc.). Three-dimensional descriptors are derived from the 3D conformation of the molecules (geometrical, as well as spatial, information of molecules, comparative molecular similarity index analysis (CoMSIA), solvent accessible area, comparative molecular field analysis (CoMFA), polar and nonpolar surface areas (PSAs and NPSAs), etc. [26]. Molecular fingerprints are another way of depicting the molecule structure where binary digits (bits) help find or differentiate between the specific substructures in the molecule. The descriptors and fingerprints are essential when studying drugs or chemicals to determine their QSAR [27].

### 2.4. Feature Selection

Feature selection involves identifying and eliminating redundant and irrelevant features to obtain significant features that can improve the accuracy of the developed models [28]. The feature selection was performed with the help of the perceptron, SVR, and DT methods in the recursive feature elimination (RFE) module available in the scikit-learn library to find the top 50, 100, 150, and 200 relevant features. Among these, the top 100 features of the perceptron method were used as input for implementing the machine learning algorithms in this study [29,30].

### 2.5. Machine Learning Algorithms

In this current study, we involved the implementation of SVM, ANN, kNN, and RF.

SVM is a supervised machine learning algorithm that can be utilized for regression and classification tasks. It generally creates several hyperplanes but needs to find the best hyperplane with a maximum margin that classifies the data more accurately. There are two categories of SVM, namely linear SVM and nonlinear SVM. Linear SVM is typically used for data that can be separated linearly, while nonlinear SVM is designed for data that cannot be separated linearly. The kernel function is also used to alter the training data with the help of which nonlinear decision surface is converted to a linear equation, i.e., usable form for data processing [19].

RF is also a supervised machine learning algorithm that can be utilized for regression and classification tasks. RF performs functions by forming decision trees using a training dataset, and the outturn it makes is based on the mean prediction [31].

An ANN is an effort to imitate the neuron network that comprises the human brain to make the computer learn things and respond accordingly as humans do. It typically comprises three layers: the input, hidden, and output layers. These layers transform the input into a meaningful output [32].

The kNN algorithm is a MLT that does not assume any specific form for the underlying data distribution and is supervised in nature. It can be applied to perform classification or regression tasks [33]. It is frequently known as memory-based, instance-based, or lazy learning. It is based on the pick out of the nearest neighbor for a query data point based on the distance, which can be calculated by Euclidean distance, Minkowski distance, Manhattan distance, Hamming distance, etc.

### 2.6. Generation of Random Datasets

To create independent validation datasets, we used a random selection process to choose approximately 10% of the available data, while the remaining 90% was utilized for training and testing purposes of the models. We repeated this procedure five times, resulting in five sets of training/testing and independent validation data, each containing 238 molecules (T^214^ + V^24^).

### 2.7. Ten-Fold Cross-Validation

To assess the performance of the machine learning predictive models, we employed the ten-fold cross-validation method. This technique involved splitting the training/testing dataset into ten equal parts. During each iteration, nine parts were combined for training, while the remaining part was used for testing to assess the model’s performance. All ten parts were used as testing data at least once, and the overall model performance was evaluated based on the average performance of all the testing parts. Additionally, to validate the performance of the developed model, we used an independent/external dataset that was not utilized during the model’s training and testing.

### 2.8. Model Performance Assessment

The developed model performance was evaluated by calculating the mean absolute error (*MAE*), mean squared error (*MSE*), root mean squared error (*RMSE*), coefficient of determination (R^2^), and Pearson’s correlation coefficient (*PCC* or R) using the formulas as given below.
(1)$$PCC=\frac{n\sum_{n=1}^{n}\;E_{i}^{act}E_{i}^{pred}-\sum_{n=1}^{n}\;E_{i}^{act}\sum_{n=1}^{n}\;E_{i}^{pred}}{\sqrt{n\sum_{n=1}^{n}\;{{\left(E_{i}^{act}\right)}^{2}-({\sum_{n=1}^{n}\;E}_{i}^{act})}^{2}}-\sqrt{n\sum_{n=1}^{n}\;{{\left(E_{i}^{pred}\right)}^{2}-({\sum_{n=1}^{n}\;E}_{i}^{pred})}^{2}}}$$
(2)$$MAE=\frac{1}{n}\sum_{n=1}^{n}\;\left|E_{i}^{pred}-E_{i}^{act}\right|$$
(3)$$MSE=\frac{1}{n}\sum_{n=1}^{n}\;\left(E_{i}^{pred}-E_{i}^{act}\right)²$$
(4)$$RMSE=\sqrt{\frac{1}{n}\sum_{n=1}^{n}\;{\left(E_{i}^{pred}-E_{i}^{act}\right)}^{2}}$$
where *n*, *Eact*, and *Epred* are the dataset size and actual and predicted values, respectively.

### 2.9. Applicability Domain Analysis

Moreover, along with the model performance, model accuracy for the new prediction also plays a crucial role. Applicability domain analysis defines the boundary of the developed model for its reliability. For accurate predictions of a new compound using a developed model, it is essential for the chemical properties of the compound to fall within the applicability domain of the compounds employed in training the model [34]. The reliability of these developed models was assessed using the William’s plot based on the distance-based leverage approach. These plots depict the relationship between the leverage and standardized residuals. The formula of the leverage threshold (*h**) is
Leverage threshold (*h**) = 3(*p* + 1)/*n*
(5)
where *p* = number of descriptors utilized in developing the model; *n* = number of compounds used in the training dataset.

The reliability of the predicted model was observed to be dependent on a majority of the data points falling within the leverage threshold (*h**). To confirm the strength and effectiveness of the developed models created using the SVM, RF, kNN, and ANN algorithms, we plotted a scatter plot between the predicted pIC50 values and actual pIC50 values.

### 2.10. Decoy Sets Analysis

Decoys were generated for these drug candidates using the DecoyFinder 2.0 tool [35]. DecoyFinder 2.0 utilizes a molecular weight-based method to generate decoys. The ZINC20 database was used as a source of a subset containing 4.78 million drug-like molecules to make the decoys [36]. Six decoy datasets were developed, having 238 random decoys of active drug candidates. Further format conversion and molecular descriptors were calculated to determine the pIC_50_ values. Eventually, a correlation was made in terms of the *PCC* between the decoy pIC_50_ and actual pIC_50_ of each decoy dataset’s equivalent active drug candidates.

### 2.11. Chemical Clustering Analysis

The chemical diversity of these drug candidates was evaluated by executing chemical clustering using ChemMine tools. We used the multidimensional scaling (MDS) algorithm and Binning clustering with the same similarity cut-off of 0.6 in both methods [37].

### 2.12. Drug Repurposing

Using the best predictive model based on SVM, we predict the potent repurposed drug candidates by scanning the more than 2000 FDA-approved drugs present within the DrugBank database [38]. We excluded those drugs from our DrugBank scanning approach that were already used in the model development. We converted the file format of these drugs and generated 17,968 molecular descriptors using PaDEL software. Further, we extracted the top 100 perceptron features involved in developing the best model. Subsequently, these DrugBank drugs, along with the 100 features, were employed to predict novel, potentially effective repurposed drug candidates with elevated pIC_50_ values against DENV.

### 2.13. Web Server Development

The best-performing SVM predictive model was implemented on the “Anti-Dengue” web server to assess the effectiveness of chemicals and drugs in inhibiting the DENV, as indicated by inhibition efficiencies such as the pIC_50_ and IC_50_ values (μM). The “Anti-Dengue” web server was constructed utilizing LAMP software (Ubuntu 12.04.2 LTS), incorporating Linux as the operating system, Apache as the web server, MySQL as the relational database management system, and PHP (Perl or Python) as the object-oriented scripting language. The front end of the “Anti-Dengue” web server was developed using HTML, CSS, and PHP, while the scripting languages, viz., python, perl, and JavaScript, were used at the back end of the web server. The web server predicts the inhibition efficiency in terms of the IC_50_ and pIC_50_ on the best-performing SVM model. To enhance user accessibility, we provide dedicated web pages such as “Help” and “Frequently Asked Questions” on the server for user guidance and assistance.

## 3. Results

### 3.1. Feature Selection Approach

Among all 17,968 descriptors, the top 100 features of the drugs were selected for developing the models. In the case of the support vector regression (SVR) method, the features are E1i, geomShape, FP258, KRFP320, KRFP307, ExtFP465, KRFPC3056, etc. Similarly, in the decision tree (DT) regression method, the features are SubFPC26, AATSC3m, ATSC1i, ATSC8p, ATSC8e, ATSC6e, ATSC6m, etc. Moreover, the perceptron method’s components are KRFPC52, ExtFP897, E3u, E2m, FP258, ExtFP41, ExtFP953, etc. The complete list of the top 100 features that were extracted using these three methods (SVR, DT, and perceptron) of the recursive feature elimination module is provided in Appendix A.

### 3.2. Performance of Developed Machine Learning-Based QSAR Models

To identify inhibitors of the DENV, we developed robust prediction models using four MLTs. These methods included SVM, ANN, kNN, and RF. The predicted models were developed using 100 top features/descriptors selected using the RFE module from the scikit-learn library.

Various statistical measures were utilized to evaluate the effectiveness of the developed QSAR models, including the *MAE*, *MSE*, *RMSE*, R^2^, and *PCC*. The *MAE*, or mean absolute error, is a metric used to measure the average magnitude of errors between the predicted and actual values. It is calculated by taking the average of the absolute differences between each predicted value and its corresponding actual value. The *MAE* tells about the closeness of the predicted values to the actual values. These values are negative-oriented values; that is, the more negative values, the more superior the developed model.

The *MSE*, or mean squared error, is a metric commonly used to quantify the average squared difference between predicted and actual values. It involves calculating the squared differences for each data point, averaging these squared differences, and then taking the square root to obtain the final result. The *MSE* gives more weight to larger errors than smaller ones, making them sensitive to outliers.

The *RMSE* measures the average magnitude of the errors between the predicted and actual values, with the square root applied to make the result more interpretable in the same units as the original data.

An R^2^ value of 1 depicts the data perfectly fitting into the model, whereas a value of 0 shows that the data do not fit into the model at all.

*PCC* values show the correlation between the inhibitors’ predicted and actual pIC50 values. *PCC* values lie between −1 and +1, where the −1 value shows a negative correlation, 0 values depict no correlation, and the +1 value implies a positive correlation. The R^2^ values show how well the data can fit in a statistical model.

The training and testing datasets for the DENV prediction models exhibited *PCC* values of 0.71 for SVM, 0.65 for ANN, 0.34 for kNN, and 0.45 for RF. For an independent dataset, the *PCC* values were 0.81 for SVM, 0.74 for ANN, 0.68 for kNN, and 0.54 for RF. The performance metrics for the best models developed using SVM, RF, kNN, and ANN for the DENV are presented in Table 1, Table 2, Table 3 and Table 4. Further information about all of the models developed for DENV inhibitors can be found in Appendix A. Detailed information on the actual and predicted IC_50_ of the independent validation dataset is available in Appendix A.

### 3.3. Applicability Domain Analysis

An applicability domain analysis using a William’s plot showed the leverage threshold (h*) value comes out to be 1.415 for models predicted using algorithms. Out of four predictive algorithms, the SVM model was found to be reliable, as most of the data points lie within the leverage threshold (h*), as given in Figure 2. Figure 3 displays a scatter plot between the actual pIC_50_ values and predicted pIC_50_ values for both the training/testing and independent validation datasets, illustrating that most of the points are clustered around the trend line. This indicates that the developed QSAR models are highly reliable. Appendix A contains the information used for the William’s plot in the applicability domain analysis. Appendix A contains information about the actual and predicted pIC_50_ values for the scatter plot.

### 3.4. Validation Using the Decoy Set

Unlike active molecules, decoys refer to molecules that cannot bind to their target. To confirm the predictive model’s reliability, the inhibitory activity in terms of the pIC50 was calculated for all six random decoy sets and then compared in terms of pIC50 with their corresponding active molecules (Appendix A). Decoy sets 1–6 showed the *PCC* values 0.117, 0.045, −0.0002, −0.091, −0.043, and −0.028, respectively, and their correlation is displayed using a scatter plot in Figure 4.

### 3.5. Chemical Diversity Analysis

A chemical diversity analysis was conducted to check the structural heterogeneity of the anti-dengue chemical compounds. A binning clustering analysis revealed that anti-dengue chemical compounds could be sorted into 124 bins or clusters (Appendix A). A 2D and 3D multidimensional scaling plot was generated to illustrate the dissimilarity of anti-dengue chemical compounds in chemical space, utilizing the same similarity cut-off as the binning clustering analysis, as shown in Figure 5.

### 3.6. Prediction of Promising Repurposed Anti-Dengue Drug Candidates

The most effective predictive model, based on SVM, was utilized to forecast repurposed drugs from the approved drugs category of the “DrugBank” database. The top 25 predicted candidates are listed in Table 5.

### 3.7. Anti-Dengue Web Server

To predict the effectiveness of anti-dengue chemicals, users should paste/upload the input in SDF format. The output will be received in a tabular format that includes Query ID, SMILES, the inhibition efficiency as pIC_50_ and IC_50_ (μM), 2D structure, and descriptor. The computation time for unknown chemicals typically ranges between 2 and 5 min. Users can keep track of their jobs by noting the job ID and accessing the “check job status” page to retrieve the results at any time. The “Anti-Dengue” web server is freely available at https://bioinfo.imtech.res.in/manojk/antidengue/.

## 4. Discussion

Dengue is an emerging health problem across the globe. Due to the absence of approved antiviral treatments or a universal vaccine for DENV infection, several research teams are focused on developing inhibitors that target various components, such as structural, nonstructural, host, and non-specific targets. In this concern, focusing on computational approaches for developing antivirals would be a better step to accelerate drug discovery research [39]. Hence, in the present research work, we developed a machine learning-based prediction algorithm, “Anti-Dengue”, to identify new potential repurposed drug candidates targeting DENV.

In this study, we employed multiple machine learning techniques (MLTs): support vector machine (SVM), artificial neural network (ANN), k-nearest neighbor (kNN), and random forest (RF) to develop a better predictive model. Additionally, we explored three feature selection methods: perceptron, SVR, and DT. By combining these MLTs with four feature sets comprising the top 50, 100, 150, and 200 features and considering five random datasets (214 molecules in training/testing and 24 molecules in independent datasets generated from 238 molecules), we developed a total of 240 models. Following an assessment of the performance parameters, such as the mean absolute error (MAE), mean squared error (MSE), root mean squared error (RMSE), coefficient of determination (R^2^), and Pearson’s correlation coefficient (PCC or R), of these models, we provided 12 predictive models details in Table 1, Table 2, Table 3 and Table 4. Finally, we selected a specific model for further analyses like the applicability domain, scatter plot, decoy dataset, etc. This chosen model is characterized by 100 features utilizing the perceptron feature selection method. Detailed information on all MLTs with 100 feature sets using all three feature selection methods and random sets is provided in Appendix A. This SVM model was integrated into the web implementation and employed to predict potential repurposed drug candidates against the dengue virus, and the top 25 predicted drug candidates are listed in Table 5.

We utilized four different MLTs, namely SVM, RF, ANN, and kNN, to develop highly effective predictive models. These MLTs have been employed by various researchers in a multitude of studies [40]. For example, Mpropred for the prediction of SARS-CoV-2 main protease antagonists [41], TargIDe for predicting the molecules with antibiofilm activity against Pseudomonas aeruginosa [42], EBOLApred for predicting cell entry inhibitors against the Ebola virus [43], and StackHCV for the identification of inhibitors against the NS5 protein of the Hepatitis C virus [44]. Similarly, we have utilized these techniques to create predictive algorithms such as AVCpred for predicting general effective antiviral compounds [15]: AVPpred, the first algorithm for predicting antiviral peptides [16], AVP-IC50 Pred for predicting antiviral peptides activity in terms of the IC_50_, i.e., the half-maximal inhibitory concentration [17], HIVprotI for predicting and designing inhibitors targeting Human Immunodeficiency Virus (HIV) proteins [18], and anti-flavi for predicting and designing various novel antiviral compounds, particularly for flaviviruses [19]. Recently, some predictive algorithms were developed for predicting repurposed drugs/inhibitors specifically for a virus, such as anti-Ebola for the Ebola virus [21] and anti-corona for SARS-CoV-2 [20]. To develop the predictor in the context of the DENV, we extracted the most relevant features from the 17,968 molecular descriptors and fingerprints. Out of all the MLTs employed to construct the predictive models, SVM outperformed RF, kNN, and ANN. SVM produced a PCC of 0.71 on the training/testing dataset and 0.81 on the independent validation dataset.

Further model robustness was cross-checked by plotting a William’s plot in the applicability domain analysis and plotting the actual vs. predicted pIC_50_ values to validate the robustness of the predicted model. We used the decoys of each active drug candidate to further check the reliability of the “Anti-Dengue” predictive models. Then, we compared the pIC_50_ values of inactive decoy molecules with their corresponding active molecule, which further confirms the reliability and robustness of the developed “Anti-Dengue” predictive models.

Furthermore, a chemical clustering analysis for the 238 molecules was also assessed using the multidimensional scaling (MDS) algorithm and binning clustering methods. Chemical clustering is generally used to identify outliers and understand chemical compounds’ arrangement in a chemical space. The binning clustering method made the chemical clusters based on the user-defined similarity cut-off values. We used a Tanimoto coefficient (Tc) of 0.6 as the similarity coefficient, which is the proportion of the features shared among two compounds divided by their union, i.e., c/(a + b + c), where c is the number of features common in both compounds, while a and b are the number of features that are unique in one or the other compound, respectively [45]. The Tanimoto coefficient value generally lies between 0 and 1, with higher values depicting greater similarity and vice versa. Using a Tc of 0.6 showed that compounds are joining with 0.6 similarity or more to aggregate numerous clusters using the “single linkage” rule. As many clusters are forming in the anti-dengue chemicals, they are well spread in the chemical space. The binning cluster results are represented in tabular form with the compound ID, bin/cluster size, and bin/cluster ID. Multidimensional scaling (MDS) creates a matrix of “item-to-item” distances, and each item is assigned with coordinates and represents these distances in the form of 2D and 3D scatter plots. MDS-generated plots show that anti-dengue chemicals are well distributed in the 2D and 3D chemical space. Binning clustering utilizes internally developed C++ implementation, and MDS uses the “cmdscale” function implemented in R. These methods showed that these chemicals are very dissimilar [20,46].

The developed predictive model identified several potentially effective repurposed drugs for the treatment of DENV from the “approved” drugs category within the DrugBank database. Furthermore, we conducted a literature review to verify the status of the top predicted drugs. We discovered that some hits have been investigated through experimental reports or in silico analysis. For example, Carro, Ana C., Luana E. Piccini, and Elsa B. Damonte tested chlorpromazine as an endocytic inhibitor against DENV-2 entry into myeloid cells in the presence or absence of antibodies [47]. Similarly, Shahen, Mohamed et al. showed that Loratadine (LRD), along with ReDuNing (RDN) and Acetaminophen, decreases the susceptibility, as well as the severity of, DENV by targeting the miRNA interacting with the potential target genes [48]. Likewise, Boonyasuppayakorn, Siwaporn et al. checked Primaquine, along with known FDA-approved antimalarial drugs like chloroquine and amodiaquine, to inhibit the viral proteases and DENV replication using protease, as well as reporter replication-based assays [49]. Malakar, Shilu et al. evaluated the four Food and Drug Administration (FDA)-approved drugs: azelaic acid, quinine sulfate, aminolevullic acid, and mitoxantrone hydrochloride. Quinine had the most potent activity against the DENV-2 virus strain. Quinine was found to inhibit DENV production by 80% compared to the controls. In a dose-dependent manner, it decreased DENV RNA and viral protein synthesis, consequently impeding replication [11]. Therefore, repurposed drug candidates predicted from our method have the potential to work as antiviral agents that could accelerate the drug discovery process for combating DENV infection.

Several researchers have conducted in silico studies aimed at identifying repurposed drugs against the DENV. These studies encompassed techniques like the transcriptomics-based bioinformatics approach, molecular simulations, molecular docking, pharmacophore model-based drug repurposing, and others [50,51]. These studies include datasets like phytocompound databases, natural products, small molecules, and FDA-approved drugs. Nonetheless, our study diverged from these methodologies, as we integrated four distinct MLTs to predict agents with anti-dengue properties. To develop the predictive models, we employed a range of chemically diverse anti-dengue compounds that have been experimentally validated by different research groups. Additionally, our best predictive models have been integrated into the web server, a feature that sets them apart from any previously documented computational studies for the DENV.

Recurring occurrences of DENV outbreaks characterized by significant mortality and fatality rates are causing significant global apprehension, as there is no approved drug or universal vaccine available for the treatment of DENV infection. Therefore, utilizing computational methods could prove highly beneficial in accelerating the discovery of potent inhibitors against the DENV. In this endeavor, “Anti-Dengue” is the first dedicated web server based on MLTs to find novel potential repurposing drug candidates against DENV infection.

The limitations of the current study are primarily associated with the size of the dataset. Specifically, the relatively small number of entries related to the dengue virus poses a constraint, as a larger dataset could enhance the predictive model’s performance. Another limitation is that the Anti-Dengue web server is currently employing a highly effective SVM-based predictive model for the identification of potential inhibitors/repurposed drugs in terms of inhibition efficiency, as indicated by the pIC_50_ and IC_50_ values (μM) against the dengue virus. Unfortunately, alternative machine learning models were not integrated due to their inferior performance on the existing dataset. It is our belief that the development of more robust predictive models using machine learning may be achievable in the future with the availability of additional data. A third limitation is that the “Anti-Dengue” web server is designed exclusively for small molecules, as it is trained on chemicals and FDA-approved drugs, and is not applicable to peptides, antibodies, etc.

## 5. Conclusions

We developed a QSAR-based algorithm, “Anti-Dengue”(https://bioinfo.imtech.res.in/manojk/antidengue/), which utilizes SVM, ANN, kNN, and RF. Predictive models were developed to identify the potent inhibitors against the DENV. The performance of these predictive models was found to be good, with a PCC of up to 0.71 on the training/testing dataset and a PCC of up to 0.81 on the independent validation dataset. Further applicability domain, chemical clustering, and decoy dataset analyses showed that these predictive models are reliable and robust in nature. The “DrugBank database” was scanned to predict the potential repurposed drug candidates against the DENV. As a result, it will facilitate the rapid development of antivirals that are effective against the DENV.

## Figures and Tables

**Figure 1 viruses-16-00045-f001:**
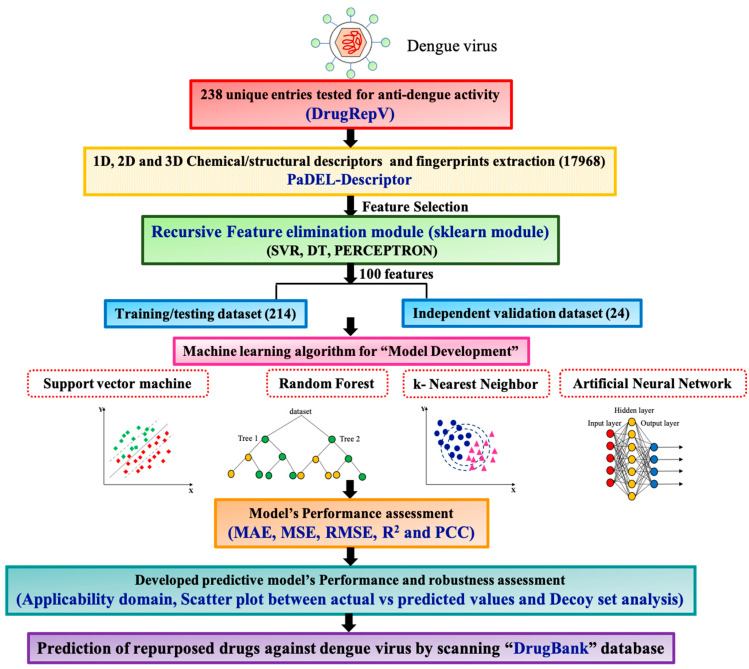
The workflow includes retrieving dengue inhibitors from DrugRepV and converting SMILES to SDF format. Molecular descriptors/fingerprints are calculated using PaDEL software, followed by the recursive feature elimination (RFE) module for feature selection. SVM, ANN, kNN, and RF MLTs are employed with ten-fold cross-validation for predictive algorithms. The performance is evaluated using *MAE*, *MSE*, *RMSE*, R^2^, and *PCC* values. Further, the model’s robustness is analyzed with applicability domain, scatter plots, and decoy sets. Potent repurposed drugs are predicted by scanning the “DrugBank” database.

**Figure 2 viruses-16-00045-f002:**
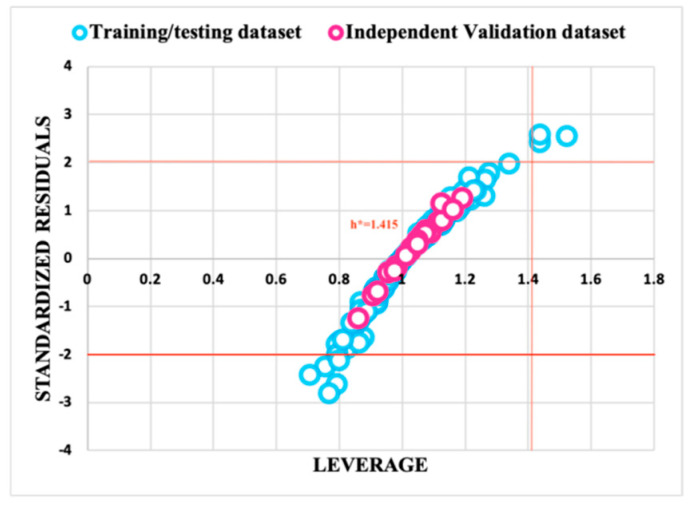
The applicability domain analysis of the support vector machine was assessed by a William’s plot between the leverage and standardized residuals of the molecules.

**Figure 3 viruses-16-00045-f003:**
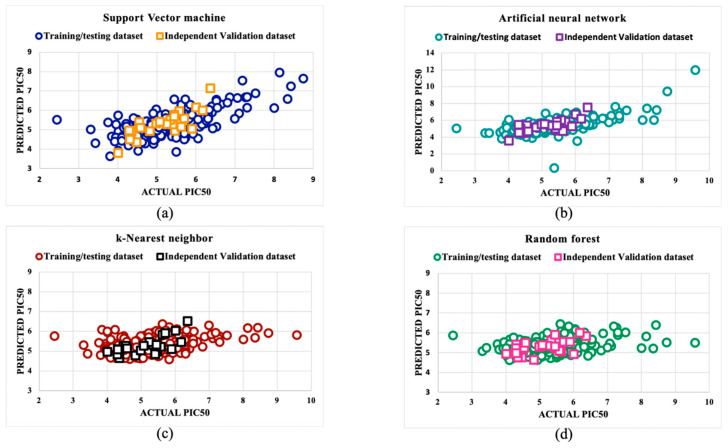
The robustness of the (**a**) support vector machine, (**b**) artificial neural network, (**c**) k-nearest neighbor, and (**d**) random forest-based predicted models was assessed by scatter plots between the actual and predicted pIC_50_ values of the molecules.

**Figure 4 viruses-16-00045-f004:**
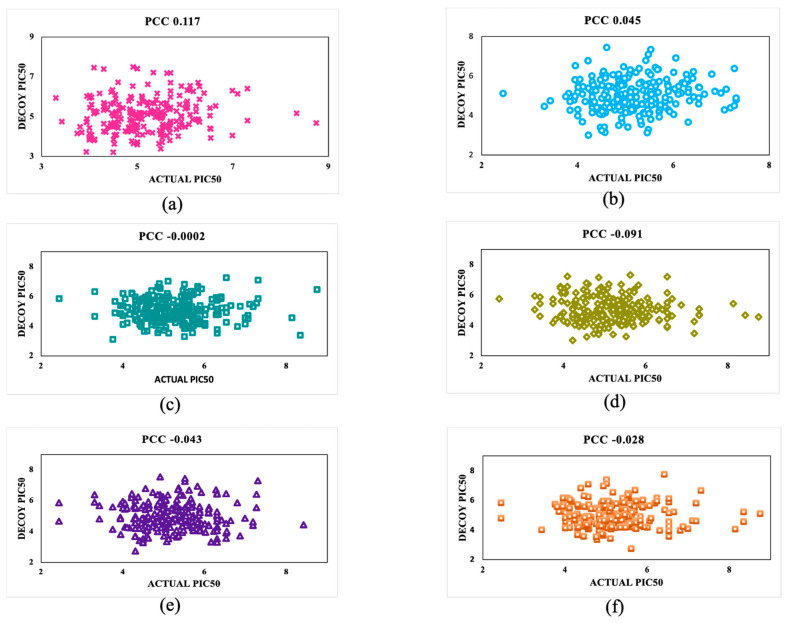
To evaluate the reliability of the predicted models based on SVM, a scatter plot was generated to compare the actual and decoy pIC50 values of (**a**) Decoy Set 1, (**b**) Decoy Set 2, (**c**) Decoy Set 3, (**d**) Decoy Set 4, (**e**) Decoy Set 5, and (**f**) Decoy Set 6.

**Figure 5 viruses-16-00045-f005:**
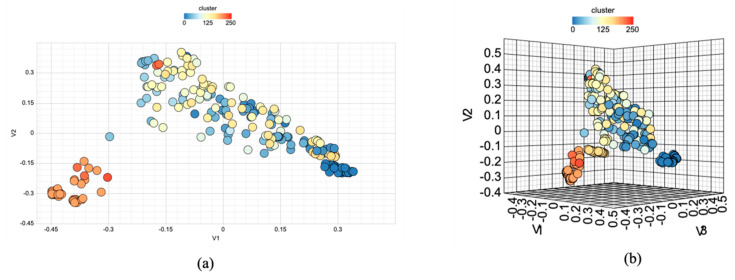
Chemical diversity analysis: (**a**) 2D multidimensional scaling plot and (**b**) 3D multidimensional scaling plot of the anti-dengue compounds.

**Table 1 viruses-16-00045-t001:** “Anti-Dengue” predictive model performances during 10-fold cross-validation using the SVM machine learning technique.

Algorithm	Feature Selection	Model Parameters	Dataset	*RMSE*	*MSE*	*MAE*	R^2^	*PCC*
**SVM**	**Perceptron**	**svm_param_32_kernel_rbf_gamma_0.005_C_10**	**T214**	**0.69**	**0.47**	**0.48**	**0.47**	**0.71**
			**V24**	**0.43**	**0.19**	**0.36**	**0.56**	**0.81**
SVM	SVR	svm_param_32_kernel_rbf_gamma_0.005_C_10	T214	0.72	0.55	0.51	0.39	0.68
			V24	0.38	0.15	0.31	0.66	0.84
SVM	DT	svm_param_1_kernel_rbf_gamma_0.1_C_0.01	T214	0.97	0.99	0.71	−0.07	0.41
			V24	0.65	0.42	0.56	0.02	0.36

**Table 2 viruses-16-00045-t002:** “Anti-Dengue” predictive model performances during 10-fold cross-validation using the ANN machine learning technique.

Algorithm	Feature Selection	Model Parameters	Dataset	*RMSE*	*MSE*	*MAE*	R^2^	*PCC*
ANN	Perceptron	ANN__paras_19_activation_identity_solver_sgd_learning_constant	T214	0.72	0.59	0.52	0.04	0.65
			V24	0.58	0.33	0.46	0.22	0.74
ANN	SVR	ANN__paras_26_activation_identity_solver_lbfgs_learning_invscaling	T214	0.52	0.32	0.36	0.62	0.67
			V24	0.34	0.11	0.26	0.74	0.90
ANN	DT	ANN__paras_14_activation_*relu*_solver_*adam*_learning_invscaling	T214	4.63	108.31	1.77	−98.1	0.45
			V24	0.9	0.82	0.63	−0.9	0.43

**Table 3 viruses-16-00045-t003:** “Anti-Dengue” predictive model performances during 10-fold cross-validation using the kNN machine learning technique.

Algorithm	Feature Selection	Model Parameters	Dataset	*RMSE*	*MSE*	*MAE*	R^2^	*PCC*
kNN	Perceptron	knn_k9	T214	0.89	0.83	0.7	0.0	0.34
			V24	0.5	0.25	0.41	0.41	0.68
kNN	SVR	knn_k7	T214	0.87	0.81	0.67	0.07	0.35
			V24	0.46	0.21	0.37	0.51	0.74
kNN	DT	knn_k9	T214	0.9	0.88	0.66	0.02	0.37
			V24	0.48	0.23	0.38	0.46	0.72

**Table 4 viruses-16-00045-t004:** “Anti-Dengue” predictive models performance during 10-fold cross-validation using RF machine learning technique.

Algorithm	Feature Selection	Model Parameters	Dataset	*RMSE*	*MSE*	*MAE*	R^2^	*PCC*
RF	Perceptron	rf__paras_30_n_200_depth_12_split_5_leaf_4	T214	0.89	0.82	0.66	0.07	0.45
			V24	0.57	0.33	0.47	0.24	0.54
RF	SVR	rf__paras_44_n_300_depth_12_split_2_leaf_2	T214	0.84	0.76	0.63	0.15	0.49
			V24	0.45	0.2	0.36	0.54	0.79
RF	DT	rf__paras_30_n_200_depth_12_split_5_leaf_4	T214	0.84	0.74	0.61	0.13	0.54
			V24	0.45	0.2	0.37	0.53	0.77

*PCC*—Pearson’s correlation coefficient, R^2^—coefficient of determination, *MAE*—mean absolute error, *MSE*—mean squared error, and *RMSE*—root mean squared error.

**Table 5 viruses-16-00045-t005:** The top hits of the predicted repurposed drug candidates.

DrugBankID	Drug Name	Primary Indication	Predicted_pIC_50_	Status
DB00014	Goserelin	Breast cancer and prostate cancer	8.42	Not yet tested
DB00644	Gonadorelin	Function of gonadotropes and the pituitary	8.19	Not yet tested
DB00666	Nafarelin	Central precocious puberty in children of both sexes and treatment of endometriosis	8.03	Not yet tested
DB11279	Brilliant green	To prevent infections of the umbilical cord	8.03	Not yet tested
DB01284	Tetracosactide	Screening of patients presumed to have adrenocortical insufficiency	7.91	Not yet tested
DB12887	Tazemetostat	Metastatic or locally advanced epithelioid sarcoma is not eligible for complete resection.	7.83	Not yet tested
DB00626	Bacitracin	Wound infections, pneumonia, skin and eye infections	7.83	Not yet tested
DB01061	Azlocillin	Pseudomonas aeruginosa, Haemophilus influenzae and Escherichia coli infections	7.81	Not yet tested
DB01403	Methotrimeprazine	For the treatment of psychosis, particular those of schizophrenia, and manic phases of bipolar disorder	7.8	Not yet tested
DB01621	Pipotiazine	Chronic non-agitated schizophrenic patients	7.67	Not yet tested
DB01147	Cloxacillin	Treatment of beta-hemolytic streptococcal, pneumococcal, and staphylococcal infections	7.67	Not yet tested
DB06788	Histrelin	Palliative treatment of advanced prostate cancer	7.65	Not yet tested
DB09320	Procaine benzylpenicillin	Local anesthetic and antibiotic combination for bacterial infections	7.62	Not yet tested
DB00434	Cyproheptadine	Appetite stimulation, allergic symptoms, and treatment of serotonin syndrome	7.51	Not yet tested
DB09570	Ixazomib	Multiple myeloma	7.51	Not yet tested
DB09473	Indium In-111 Oxyquinoline	Radiolabeling autologous leukocytes	7.5	Not yet tested
DB04826	Thenalidine	Not available	7.41	Not yet tested
DB00477	Chlorpromazine	Preoperative anxiety, nausea, vomiting, bipolar disorder, and schizophrenia	7.27	Experimental
DB00948	Mezlocillin	Lungs, urinary tract, skin gram-negative infections	7.39	Not yet tested
DB01201	Rifapentine	Pulmonary tuberculosis	7.39	Not yet tested
DB00455	Loratadine	Manage the symptoms of allergic rhinitis	6.8	Experimental
DB01087	Primaquine	To prevent relapse of vivax Malaria	6.69	Experimental
DB00468	Quinine	Uncomplicated Plasmodium falciparum Malaria	6.65	Experimental
DB01583	Liotrix	Primary, secondary or tertiary hypothyroidism	6.63	Not yet tested
DB09225	Zotepine	Schizophrenia	6.63	Not yet tested

## Data Availability

Available at https://bioinfo.imtech.res.in/manojk/antidengue/.

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
