# Peer review of "Anti-Dengue: A Machine Learning-Assisted Prediction of Small Molecule Antivirals against Dengue Virus and Implications in Drug Repurposing"

_viruses, 2023, doi:10.3390/v16010045_

Round 1

Reviewer 1 Report

Comments and Suggestions for Authors

This paper presents a computational pipeline that uses machine learning to predict drug repositioning hints for the dengue virus. The paper is well-written and appears to be technically sound. However, the authors need to clarify issues and address concerns, as follows:

1. The reader must dig deep into the paper (the Results section) to find precisely the attribute/parameter the authors predict. Right from the Introduction and Methods sections, they need to state with clarity that they predict pIC50.

2. What exactly are the entries in the anti-dengue activity DrugRepV? The authors mention 'inhibitors,' but they must be more specific. Are these so-called inhibitors drugs (small molecule type)? If not, what are they?

3. If the 'inhibitors' are drugs, why do the authors need to scan DrugBank? Just for confirmation? This aspect must be clarified. 

4. What is the difference between the computational pipeline used in this paper and the ones presented in references [21] and [22]? At first glance, it looks like the only difference is that they target different diseases; however, whether that is true or not, the authors must explain this aspect with clarity.   

5. The paper does not justify the four regression models the authors test. Why these 4 and not others?

6. The authors present the RMSE as a metric for prediction performance, but they do not use it. Why?

7. Anyway, the numbers presented by the authors for MAE, PCC, and R^2 indicate relatively poor results. Moreover, for the best model (i.e., SVM), the prediction performance is better on the validation test than on the training set; this suggests that the model is underfitting, thus calling for a more complex learning model. This further stresses the need to justify the models authors test in this paper.    

8. Do we have a ground truth for validating the repositioning hints? The paper suggests that we have literature titles and various in silico models that can be used to (at least) support these hints. However, the authors present these supporting arguments only as anecdotal evidence, which is good but not sufficient. Can they present some statistics to validate/support these hints? For example, what percentage of the top hints are supported by literature, what percentage is supported by in silico methods, and what percentage is not supported by other types of information? 

Comments on the Quality of English Language

English is fine.

Reviewer 2 Report

Comments and Suggestions for Authors

In the current study, authors claimed that they developed "Anti-dengue” algorithm predicting dengue virus inhibitors using quantitative structure-activity relationship 13 (QSAR) and MLTs. The parameters chosen in this study do not look adequate for claiming potential drug candidates that could be repurposed. The link given also could not be opened for checking. The figures and supplementary tables are also not convincing for this algorithm is worthful for future anti-dengue studies. Authors also repeat themselves in results, discussion and conclusion parts. They also claimed that drug repurposing could be a promising approach to looking for effective antivirals against the dengue virus based on some well-known drugs but they do not have adequate evidence for these claims. Overall, I strongly recommend reject of this manuscript.

Comments on the Quality of English Language

Moderate editing of English language required.

Reviewer 3 Report

Comments and Suggestions for Authors

The study titled "Machine Learning-Assisted Prediction of Small Molecule Antivirals against Dengue Virus and Implications in Drug Repurposing" by Gautam et al. offers valuable insights into drug repurposing for Dengue Virus, a virus that causes persistent epidemics in tropical and subtropical regions globally. Despite the prevalence of Dengue, there are currently no approved antiviral treatments available, making therapeutic development a critical need. In response to this challenge, the authors developed the "Anti-dengue" algorithm, which leverages quantitative structure-activity relationship (QSAR) and machine learning techniques (MLTs) to predict potential inhibitors of the Dengue Virus.  The "DrugRepV" database was employed to extract repurposed drugs along with their corresponding IC50 values. Molecular descriptors and fingerprints were calculated using the PaDEL software. Subsequently, the dataset was divided into training/testing and independent validation sets. The authors constructed regression-based predictive models through a 10-fold cross-validation process, employing various machine learning approaches, such as Support Vector Machine (SVM), Artificial Neural Network (ANN), k-Nearest Neighbors (kNN), and Random Forest (RF). The best-performing model demonstrated a Pearson Correlation Coefficient (PCC) of 0.71 on the training/testing dataset and 0.81 on the independent validation dataset. The reliability and robustness of the model were assessed through William's plot, scatter plot, decoy set, and chemical clustering analyses. These predictive models were then utilized to identify potential drug candidates for repurposing, resulting in the identification of bortezomib, goserelin, gonadorelin, and nafarelin as potential repurposed drugs with high pIC50 values. The "Anti-dengue" algorithm and webserver hold promise in expediting the development of antiviral drugs against Dengue Virus, and the authors have made the Anti-dengue webserver freely accessible at https://bioinfo.imtech.res.in/manojk/antidengue/.

While I commend the authors for their dedicated work, several important points require clarification:

1. It would be helpful to specify which species of the Dengue Virus the study focused on, as there are multiple species.

2. The study should provide more details about the nature of the selected datasets, particularly whether they target viral load or specific viral proteins/enzymes.

3. Clarify what the optimal IC50 value is for the selection of candidate drugs included in the training set.

4. The term "Format conversion" should be corrected to "Descriptor calculation" for accuracy.

5. The methodology for webserver development should be explained in greater detail, including accessibility and any issues that may currently impede its review. In accessible:https://bioinfo.imtech.res.in/manojk/antidengue/.

6. Consider addressing whether the authors explored the use of the XGBoost algorithm, which is known for producing more robust bioactivity-related models.

7. It is important to outline the limitations of the current model to provide a comprehensive view of its applicability and potential constraints.

Comments on the Quality of English Language

 Minor editing of English language required

Reviewer 4 Report

Comments and Suggestions for Authors

This paper presents the development of the "Anti-dengue" algorithm that uses machine learning techniques to identify potential repurposed drug candidates against the dengue virus, demonstrating good predictive performance and providing a webserver for public use. Due to the interest of the topic that it addresses, I find the work of utility for the scientific community. In this sense, I think that it could be suitable for publication in the Viruses journal provided that the following comments are implemented within the document: 

- Besides the PCC, the paper could include additional statistical analyses such as the root mean square error (RMSE), or mean absolute error (MAE) for regression tasks to provide a more comprehensive understanding of the model's performance.

- While the identification of potential drugs is significant, experimental validation of these drugs against the dengue virus would significantly strengthen the paper's conclusions. In any case, in vitro or in vivo studies to confirm the efficacy of the predicted inhibitors could be suggested.

- Provide more detailed information on the software implementation, computational requirements, and potential limitations of the "Anti-dengue" webserver, as this would help researchers with varying resources.

- In the same vein, a thorough discussion of the limitations of the current study is always beneficial. This includes potential biases in the DrugRepV database, limitations of the machine learning models used, and how the model might perform with structurally diverse sets of inhibitors.

- Provide information on the sustainability and maintenance plans for the "Anti-dengue" webserver. Users will be interested to know how long the server will be supported and updated.

Comments on the Quality of English Language

-

Round 2

Reviewer 1 Report

Comments and Suggestions for Authors

The authors addressed my concerns.

Comments on the Quality of English Language

No major issues detected.

Reviewer 2 Report

Comments and Suggestions for Authors

Authors claimed that they employed multiple Machine Learning Techniques (MLTs): Support Vector Machine (SVM), Artificial Neural Network (ANN), k Nearest Neighbor (kNN), and Random Forest (RF) to develop a better predictive model. In revised manuscript, they responded questions well and they generally presented the system well. However, when you enter the system from the link that they gave, there is not anything useful. The link they given is open now. However, in the website there is not any information, guidance to how to use. There is only a box, which you can paste your structure(s) in SDF format. Then, when you follow submitting, there are a lot of numbers of which it is not understandable what they mean. I am still not convinced this system is totally reliable and applicable.

The good points are updated figures and supplementary tables included in the manuscript. They also expanded the discussion part well. Authors should update the online system to make it clearer and more understandable for users. I recommend “minor revision”.
